# Clinical Outcomes for Nasopharyngeal Cancer in Non-Asian Patients: A Single-Center Experience

**DOI:** 10.3390/jcm14041177

**Published:** 2025-02-11

**Authors:** Renata Zahu, Daniela Urian, Vlad Manolescu, Andrei Ungureanu, Carmen Bodale, Alexandru Iacob, Stefan Cristian Vesa, Cristina Tiple, Gabriel Kacso

**Affiliations:** 1Department of Oncology and Radiotherapy, “Iuliu Hațieganu” University of Medicine and Pharmacy, 400347 Cluj-Napoca, Romania; gabi.kacso@gmail.com; 2Amethyst Radiotherapy Center Cluj, 407280 Florești, Romania; daniela.sturzu@amethyst-radiotherapy.com (D.U.); vlad.manolescu@amethyst-radiotherapy.com (V.M.); andrei.ungureanu@amethyst-radiotherapy.com (A.U.); carmen.bodale@amethyst-radiotherapy.com (C.B.); alexandru.iacob@amethyst-radiotherapy.com (A.I.); 3Department 2 Functional Sciences, Discipline of Pharmacology, Toxicology and Clinical Pharmacology, Faculty of Medicine, “Iuliu Hațieganu” University of Medicine and Pharmacy, 400337 Cluj-Napoca, Romania; stefan.vesa@umfcluj.ro; 4Department of ENT, “Iuliu Hațieganu” University of Medicine and Pharmacy, 400337 Cluj-Napoca, Romania; cristina.tiple@umfcluj.ro

**Keywords:** nasopharynx cancer, radiotherapy, chemotherapy

## Abstract

**Background/Objectives**: According to Globocan, Romania has the highest incidence of nasopharyngeal cancer (NPC) in Europe. Our objective was to evaluate the survival data for a cohort of non-Asian patient population treated with curative intent at a tertiary cancer center in Romania. **Methods**: We retrospectively analyzed 161 patients with histologically proven, non-metastatic NPC treated at our institution between October 2014 and December 2021 with intensity modulated arc radiotherapy (IMRT) with or without neoadjuvant or concomitant chemotherapy according to the stage of the disease. Kaplan-Meier estimates of overall, disease-free, locoregional relapse free and distant metastasis free survival were calculated. The log-rank test was used to determine significant prognostic determinants of overall and disease-free survival. **Results**: The median age was 50 years (range 19–80), 88% had nonkeratinizing undifferentiated carcinoma. Epstein Barr virus status was not evaluated routinely. 42.2% of patients were stage III and 46% stage IVA disease. Induction chemotherapy was prescribed for 72.7% of patients and 89.4% received concurrent chemotherapy. After a median follow up of 44 months (range: 3.6, 104.7 months), the estimated 3 years overall, disease free, locoregional relapse free and distant metastasis free survival of the entire cohort were 82.6%, 73.3%, 83.2% and 86.3% respectively. On testing interactions, concomitant chemotherapy offered significant survival benefit (HR—0.287; 95% CI 0.137–0.603; *p* = 0.001) and cumulative Cisplatin dose of more than 100 mg/mp was statistically significant for survival (HR—0.350;95% CI 0.157–0.779; *p* = 0.01) **Conclusions**: This is the largest retrospective series of nasopharyngeal cancer from Romania reporting survival data. Despite the high percentage of advanced stage disease our data shows very good disease control. Compliance to optimal concomitant chemotherapy should represent a priority in clinical practice in a non-Asian patient population.

## 1. Introduction

Nasopharyngeal carcinoma (NPC) is endemic in South-East Asia and North Africa but is considered a rare head and neck cancer in the rest of the world with an incidence lower than 1/100, 000 [1]. In the last two decades an increasing trend was observed in the age-standardized incidence rates globally, Romania showing one of the highest percentage changes, with almost 200% increase. According to Globocan, Romania had the highest incidence for nasopharyngeal cancer in Europe in 2022 with an estimated 358 new cases, with an age standardized rate of 1.3/100,000 [2], unfortunately along with an increase of almost 125% in the age-standardized mortality rates in the last two decades [3]. Differences in the incidence of NPC might be also due to genetics, ethnicities, environmental factors, lifestyle, diet with preserved foods and salted fish [1,4]. Ethnicity plays an important role in prognosis, with publications showing that Asians and African Americans have better disease specific survival than Caucasian patients [5] and that chemotherapy added to radiotherapy(RT) was more beneficial in non-Asian patients [6].

The standard treatment is high dose RT combined with chemotherapy established by the landmark Intergroup 00-99 trial [7]. Most of the newer randomized trials regarding induction chemotherapy or treatment deintensification are coming from Asian countries [8]. Intensity modulated radiotherapy(IMRT) is the preferred radiotherapy technique because it results in less late complications and a better quality of life [9].

Our objective was to evaluate the survival data for a cohort of non-Asian, Caucasian patient population treated with IMRT at an academic cancer center in one of the most affected European countries. We also analyzed the compliance and impact of concomitant chemotherapy on the treatment results.

## 2. Materials and Methods

We retrospectively analyzed adult patients over 18 years old, with histologically proven, non-metastatic NPC treated with curative intent at our institution between October 2014 and December 2021. We have excluded metastatic patients or those who were not treated with curative intent. Staging procedures included MRI of the head and neck and CT Thorax and Abdomen and in equivocal cases PET/CT. All patients were restaged according to AJCC/UICC 8th edition, and all patients were treated with high dose intensity modulated radiotherapy (IMRT) with image guidance with or without neoadjuvant or concomitant chemotherapy according to the stage of the disease. EBV testing was not performed routinely. All treatments were according to institutional protocols directly adapted from international guidelines. Induction chemotherapy was conducted every three weeks with Cisplatin or Carboplatin containing doublets either with Gemcitabine or Epirubicine or triplets including 5FU and Paclitaxel. Radical IMRT was delivered with 70 Gy in 33 or 35 fractions to the gross tumor volume (GTV), involved nodes and high-risk clinical target volume (CTV HR), 60–63 Gy to the intermediate risk clinical target volume (CTV IR) and 54–56 Gy to low-risk elective volumes (CTV LR).Planning target volume (PTV) margins of 3 mm were used. Concomitant chemotherapy consisted of weekly Cisplatin 40 mg/m^2^, Cisplatin 100 mg/m^2^ every three weeks or weekly Carboplatin AUC 2 in selected cases with contraindication to Cisplatin. Acute toxicity was graded according to the Common Toxicity Criteria for Adverse Events version 4.0 (CTCAE 4.0). End of treatment PET/CT scan was ordered three months after RT. Patients were regularly followed up every 3 months in the first 2 years by the treating radiation oncologist alternatively with a head and neck specialist, every 6 months in year 3 and annually thereafter with flexible fibroscopy, CT and MRI imaging. We used the Common Terminology Criteria for Adverse Events, version 4.0, to grade acute toxic effects during treatment and late toxic effects that were associated with radiotherapy were graded according to the Late Radiation Morbidity Scoring Scheme of the Radiation Therapy Oncology Group [10]. Disease free survival (DFS) was defined as the date from RT initiation to the first documented disease progression or last follow up. Overall survival (OS), locoregional relapse free survival (LRRFS) and distant metastasis free survival (DMFS) were defined as the date from RT initiation to death, first locoregional relapse and first metastasis or last follow up respectively. We used MedCalc^®^ Statistical Software version 22.021 (MedCalc Software Ltd., Ostend, Belgium; https://www.medcalc.org; 1 December 2024) for statistical analysis. Treatment outcomes were analyzed in relation to patient and tumor characteristics using univariate and multivariate analyses. T category was divided into T1–2 and T3–4 groups. N category was divided into N0–1 and N2–3 groups. AJCC stage was divided into stage I–II and stage III–IVA groups. The estimate of OS, DFS, LRRFS, DMFS was calculated actuarially with the Kaplan–Meier method. Comparison between groups was done with the k sample log-rank test. Multivariate analysis was performed using the Cox proportional hazards model to define independent predictors among potential prognostic factors. Differences between groups were calculated using the χ^2^ test for categorical variables. All *p* values were two-sided, and *p* < 0.05 was considered as the statistical significance limit.

## 3. Results

A total of 184 consecutive patients with nasopharyngeal cancer were addressed to our institution in the study timeframe. We excluded 7 patients staged IVB with distant metastases at the time of the diagnosis, 10 patients with reirradiation for local or locoregional disease and 6 pediatric patients. Therefore, the final analysis included 161 patients treated with curative intent. Their baseline characteristics are presented in Table 1. The median age at diagnosis was 50 years (range 19, 80), 67.1% were male. Most of the patients, 42.2% were stage III and 46% stage IVA disease. 142 patients had WHO type III, undifferentiated carcinoma and only two cases had keratinizing squamous cell carcinoma. Induction chemotherapy was delivered for 72.7% of patients: Gemcitabine with Cisplatin (26.5%), 5FU with Cisplatin (26.5%), Paclitaxel, 5FU with Cisplatin (19.7%), Epirubicine with Cisplatin (19.7%). Of those receiving induction chemotherapy 99.1% completed at least 2 cycles and 71% 3 cycles. The mean number of cycles administered was 2.8. All patients completed intensity modulated radiotherapy with conventional fractionation 70 Gy in 33 or 35 fractions with simultaneous integrated boost. Concurrent chemotherapy was administered to 89.4% of patients. 101 patients received weekly Cisplatin 40 mg/m^2^, 36 patients high dose Cisplatin and only 7 cases received weekly Carboplatin together with radiotherapy. The mean cumulated Cisplatin dose was 175 mg/m^2^ for the entire cohort, 164 mg/m^2^ in the weekly arm and 205 mg/m^2^ in the 3-weekly arm.

### 3.1. Toxicity

Except one patient who died during radiotherapy because of an acute respiratory infection, all patients completed radiotherapy. Acute toxicities during concomitant chemoradiation were recorded and are presented in Table 2. Toxicities during induction chemotherapy were not available for analysis. Only severe (grade 3 and 4) late effects were recorded during follow up. Five patients developed grade 3 neck fibrosis, one of them known with dermatomyositis, another one after adjuvant reirradiation for relapsed neck node salvaged with surgery. One patient developed grade 1 visual disturbance, initially with T4 disease with cavernous sinus involvement. One patient developed cranial nerve IX and X palsy, and one patient died because of a fatal epistaxis one year after radiotherapy, related on autopsy to a local relapse not visualized on routine imaging.

### 3.2. Survival

The median follow up was 44 months (range 3.6, 104.7). At the time of the analysis 116 patients were alive. Six patients died unrelated to their nasopharyngeal cancer diagnosis. The cause of death was metachronous digestive cancer in two patients, uncontrolled infections in other two, heart attack in one case and neurodegenerative disease in another one. One treatment related death was registered during chemoradiotherapy, due to a respiratory infection. Thirty-eight patients died because of their progressive disease. In this study, 17 patients developed locoregional recurrence. Three of them were salvaged with lymphadenectomy and reirradiation. A total of 25 patients developed distant metastases during the follow-up period. The most common site was bone and liver. The 3-year OS, DFS, LRRFS and DMFS rate of the entire cohort of patients was 82.6%, 73.3%, 83.2% and 86.3% respectively. Survival curves are depicted in Figure 1. Univariate analysis revealed that age, T stage and concomitant chemotherapy were significant prognostic factors. Data are presented in Table 3. On testing interactions, concomitant chemotherapy offered significant survival benefit (HR—0.287 (95% CI 0.137–0.603); *p* = 0.001) when compared to RT alone. There was no difference between weekly and three-weekly Cisplatin administration. (−0.510 (95% CI 0.147–1.762); *p* = 0.287) Cumulative Cisplatin dose of at least 100 mg/m^2^ was statistically significant for survival (HR—0.350 (95%CI 0.157–0.779); *p* = 0.01). Survival according to concomitant chemotherapy and Cisplatin administration type is depicted in Figure 2.

## 4. Discussion

This retrospective study describes the largest reported single center series of nasopharyngeal cancer treated in Romania, the country with the highest incidence in Europe [2]. Nasopharyngeal cancer is known to have the highest age standardized incidence risk (ASIR) in Asia, 3 per 100.000 persons-years in China but in some cantons reaching 25. While the ASIR is 0.43 in Central and Eastern Europe, Romania’s ASIR is 3 times higher [11].

Epstein Bar Virus (EBV) is reported to be associated with NPC risk. EBV infects nearly 90% of the population but despite this high number, the global incidence of NPC is relatively low. This suggests that other factors like genetic predisposition, environmental factors and diet like salted fish, salted vegetables and preserved food may play a role in carcinogenesis [4]. Fountzilas et al. investigated the mutational profile in 93 Greek and 34 Romanian patients and they found 89% EBV positivity [12]. Other two small immunohistochemical reports studied EBV positivity in Romanian nasopharyngeal cancer patients and reported conflicting results. In one of these studies only 2 cases out of 36 tested were positive for EBV, in the other study all 19 undifferentiated NPC were EBV positive [13,14]. There are no published studies about dietary habits or other risk factors from this region. Undifferentiated subtypes are associated with EBV prevalence in endemic areas and the squamous keratinizing subtype with smoking and alcohol consumption in the nonendemic region. In our study 98.2% were Type 2a and 2b and only two cases had squamous keratinizing type, both active smokers. Unfortunately, routine EBV testing was not done in our patient cohort because of the lack of insurance coverage for DNA or antibody testing. EBV DNA copy number is an important prognostic factor and there are studies undergoing to adapt treatment intensity based on EBV level [15,16,17]. Zhang et al. suggests that patients with a low pretreatment cell-free Epstein-Barr virus DNA load (<4000 copies/mL) might not benefit from induction chemotherapy (5-year OS, 90.6% *v* 91.4%, *p* = 0.77) [18]. In a multicentric study from nonendemic countries, Bossi et al. found that EBV negative patients did not benefit from intensive regimens including induction or adjuvant chemotherapy. They suggest that these patients should be treated as other head and neck cancer sites, only with concomitant chemoradiotherapy [19].

NPC susceptibility was connected also to HLA genes polymorphism; HLA-A, -B, -C, -DRB1, and -DQB1 loci were determined as common and well-documented alleles in Chinese population. In north African countries like Tunisia and Morocco, HLA-A10 and HLA-B18 were reported to have higher frequency. Similar studies would be needed for the Romanian patient population [20,21,22].

Some new evidence suggests that Human Papilloma Virus infection may play a role in pathogenesis of NPC similar to oropharyngeal cancer [23]. In a SEER database retrospective study of 517 NPC cases who had known HPV testing, 180 patients (34.8%) had HPV-positive NPC but with no effect on prognosis [24]. In the study of Ruuskanen from a low incidence country, HPV positivity represented only 14% and no coinfection with EBV was observed. Marin et al. reported only 2 cases (5.5%) of p16 positive NPC in their series [25]. In depth study of HPV positivity would be of high value since the region is also known for the highest incidence for cervical cancer and lack of systematic HPV vaccination.

TNM staging predicts prognosis and guides treatment selection. Our patient population was dominated by very advanced stage disease, 80.1% of patients having N2 or N3 lymph node involvement and 88.2% stage III and IV, higher compared to a multicenter study from Turkey reporting on 563 patients, where neck nodes were staged N2 or N3 in 67% of cases and there were 78.7% stages III and IVa disease [26]. In other nonendemic patient series 74% of the patients were stage III and IVa [27]. Compared to this, the Hong Kong Cancer Registry data base, analyzing 3328 NPC patients shows an incidence of 63.8% for stage III and IVA, explained by the higher awareness and diagnostic capability in endemic regions [28].

In our study 72.7% of the patients received induction chemotherapy with platinum doublets or triplets and no patient was given adjuvant chemotherapy. Those who didn’t receive induction chemotherapy were patients with stage I, II and selected stage III patients or those presenting contraindication to chemotherapy. The reason for not using adjuvant chemotherapy is because it is difficult to execute after high dose radiotherapy. Chen reported that approximately 40% of patients could not complete the prescribed adjuvant treatment cycles and reduction of the planned dose was required [29]. The standard regimen for induction is a combination of Gemcitabine plus Cisplatin or Docetaxel, 5FU and Cisplatin (TPF). TPF offers a slight improvement in disease control over Cisplatin+ 5FU when followed by concomitant chemoradiotherapy [30]. Zhang reported an impressive compliance to induction chemotherapy, 96.7% of the enrolled patients complete 3 cycles of induction [31]. In our cohort the compliance rate to 3 cycles induction chemotherapy was only 71%. Unfortunately, comprehensive toxicity data reported during induction chemotherapy was not available.

Concomitant chemoradiotherapy with the addition of induction chemotherapy is the standard of care treatment in locally advanced stages following the results of several trials from endemic regions [18,32].

A meta-analysis which included 28 trials and 8214 patients showed that both induction chemotherapy and adjuvant chemotherapy were superior to chemoradiation (CRT) alone, but induction chemotherapy was associated with greater benefit for distant progression (HR, 0.66; 95% CI, 0.47–0.93) [33,34]. However, the Hellenic Cooperative Oncology Groups’s trial conducted in a non-Asian population found no benefit in overall survival by adding induction Cisplatin, Paclitaxel and Epirubicin to concomitant CRT in 141 patients [35]. Another retrospective study from France reported no superior survival with induction chemotherapy [36]. In our study induction chemotherapy was not significant for overall survival. Prospective, randomized trials are needed to clarify the role of induction chemotherapy in nonendemic patients.

Concomitant CRT significantly improved overall survival (hazard ratio [HR] 0·79, 95% CI 0·73–0·86, *p* < 0·0001) with an absolute benefit at 5 years 6·3%, 95% CI 3·5–9·1 in the MACH NPC analysis of 19 trials and 4806 patients [37]. The Chinese Cancer Association guidelines recommends to achieve a cumulative concomitant Cisplatin dose of at least 200 mg/m^2^ [38]. Our cumulative Cisplatin dose reached 175 mg/m^2^. Several studies demonstrated that cumulative Cisplatin dose was an independent prognostic factor for outcome but the cut-off value for the optimal cumulative dose is still controversial. Ou et al. showed a prognostic value for 300 mg/m^2^, while Peng et al. recommend 240 mg/m^2^ [39,40]. In our study there was a clear benefit for a cumulative Cisplatin dose of at least 100 mg/m^2^.

IMRT represents the standard of care in terms of RT technique. A meta-analysis of 8 studies with 3570 patients showed improved 5 year locoregional control (OR 1.94 [95% CI 1·53–2·46]) and overall survival (1·51 [95% CI 1·23–1·87]) compared to 2D and 3D conformal radiotherapy [41]. For many years intensity modulated radiotherapy was not available in the country and patients were treated with conventional 2D or 3D conformal radiotherapy. Our study is the first to report on the results of IMRT introduced in routine clinical practice with 99% compliance rate to the prescribed doses. Despite the high percentage of advanced stage disease, 88.2% stage III and IVa, our data shows 82.6% overall survival at 3 years. This is comparable with data reported from other nonendemic area [26,27].

Dysphagia, xerostomia, aspiration pneumonia, trismus, hypoglossal nerve dysfunction and temporal lobe necrosis are considered late complications induced by radiotherapy [42]. Radiation induced fibrosis and microvascular injury with secondary ischemia are the main underlying pathological mechanisms. Increased high dose radiation volumes are associated with higher risk of late complications, especially chronic dysphagia [43]. In our cohort we have observed mainly neck fibrosis and trismus and only one cranial nerve palsy. With a median of 44 months follow up time, some late side effects might not be diagnosed and a longer follow up would give a better evaluation of radiation induced complications. Survivors should benefit from specialized clinical visits dedicated to assess late complications. This could result in a more comprehensive evaluation of late toxicity and could assist improvements in radiotherapy techniques.

We acknowledge the limitations of the present study consisting mainly of its retrospective single-centered nature and the lack of EBV testing. We presented the outcome only for patients treated with curative intent RT and did not include metastatic or locally relapsed patients treated with reirradiation. 

## 5. Conclusions

We present a large-scale, single center retrospective analysis of NPC outcomes in the era of IMRT from Romania, the country with the highest incidence in Europe. Survival, locoregional and distant control rates are similar to other reported data in a population with high percentage of advanced disease. Systematic EBV and HPV testing, better awareness in diagnosis, multicenter collaboration and quality assurance programs for treatment across the country should represent important practice points for the future. The role of induction chemotherapy in non-endemic population needs further evaluation in randomized clinical trials.

## Figures and Tables

**Figure 1 jcm-14-01177-f001:**
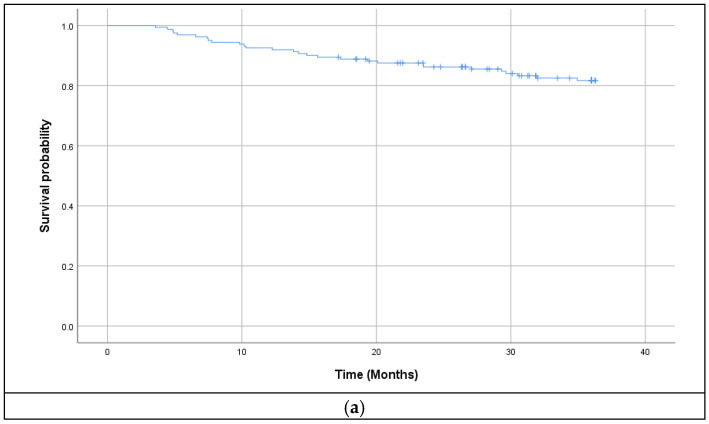
(**a**) Overall survival; (**b**) Disease free survival; (**c**) Locoregional relapse free survival; (**d**) Distant metastasis free survival.

**Figure 2 jcm-14-01177-f002:**
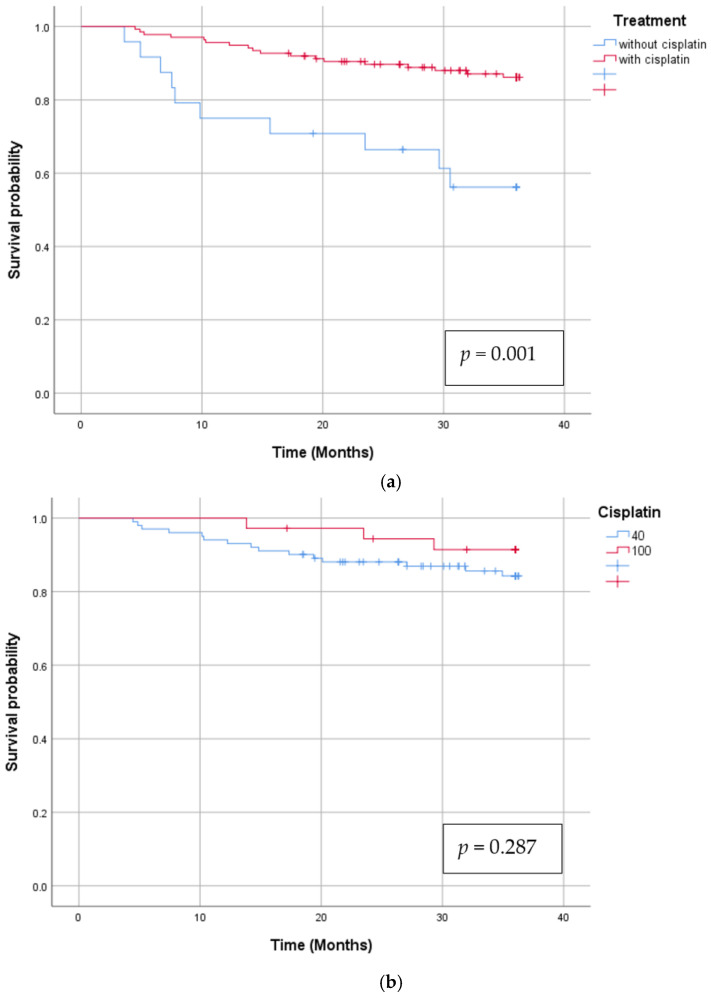
Overall survival according to Cisplatin administration (**a**) Concomitant Cisplatin versus No Cisplatin (**b**). High dose, three weekly Cisplatin versus weekly Cisplatin.

**Table 1 jcm-14-01177-t001:** Basic clinicopathological characteristics.

Characteristic	Number (%)
**Age, years**	
Median	50 (19.80)
<45	52 (32.3)
>45	109 (67.7)
**Gender**	
Female	53 (32.9)
Male	108 (67.1)
**Pathological type**	
WHO type I	2 (1.2)
WHO type II	17 (10.6)
WHO type III	142 (88.2)
**T stage**	
T1	45 (28)
T2	50 (31.1)
T3	20 (12.4)
T4	46 (28.6)
**N stage**	
N0	12 (7.5)
N1	20 (12.4)
N2	94 (58.4)
N3	35 (21.7)
**AJCC/UICC 8th ed Stage, all Mo**	
I	4 (2.5)
II	15 (9.3)
III	68 (42.2)
IVA	74 (46)

**Table 2 jcm-14-01177-t002:** Acute toxicities recorded during chemoradiation.

	Mucositis (%)	Dermatitis (%)	Neutropenia (%)	Anemia (%)	Thrombocytopenia (%)
G0	0	0	118 (73.3%)	73 (45.3)	116 (72)
G1	43 (26.7)	120 (74.5)	29 (18)	74 (46)	30 (18.6)
G2	90 (55.9)	36 (22.4)	12 (7.5)	14 (8.7)	15 (9.3)
G3	28 (17.4)	5 (3.1)	2 (1.2)	0	0
G4	0	0	0	0	0

**Table 3 jcm-14-01177-t003:** Univariate analysis of prognostic factors.

	*p* Value	HR	95% CI	
Overall survival Age T1,T2 vs. T3,T4 N0,N1 vs. N2,N3 Induction chemo Concomitant chemo Cisplatin 40 mg/ m^2^ vs. 100 mg/ m^2^	0.001 0.028 0.690 0.700 0.001 0.287	4.029 0.918 1.213 1.179 0.232 0.510	1.914 0.430 0.461 0.501 0.102 0.147	8.479 1.960 3.191 2.774 0.528 1.762
Disease free survival Age T1,T2 vs. T3,T4 N0,N1 vs. N2,N3 Induction chemo Concomitant chemo Cisplatin 40 mg/m^2^ vs. 100 mg/m^2^	0.011 0.634 0.251 0.305 0.082 0.825	2.289 1.157 1.657 1.469 0.487 0.914	1.209 0.634 0.699 0.705 0.216 0.410	4.334 2.113 3.927 3.064 1.095 2.034
Locoregional relapse free survival Age T1,T2 vs. T3,T4 N0,N1 vs. N2,N3 Induction chemo Concomitant chemo Cisplatin 40 mg/m^2^ vs. 100 mg/m^2^	0.033 0.676 0.701 0.780 0.234 0.349	2.387 1.176 1.210 1.130 0.234 0.349	1.071 0.550 0.458 0.478 0.181 0.202	5.317 2.512 3.197 2.675 1.519 1.761
Distant metastases free survival Age T1,T2 vs. T3,T4 N0,N1 vs. N2,N3 Induction chemo Concomitant chemo Cisplatin 40 mg/m^2^ vs. 100 mg/m^2^	0.098 0.994 0.187 0.304 0.029 0.435	2.133 0.997 2.661 1.766 0.329 1.545	0.868 0.426 0.622 0.597 0.121 0.518	5.240 2.332 11.385 5.218 0.895 4.611

## Data Availability

The data presented in this study is available on request from the corresponding author.

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
