# Peer review of "Clinical Outcomes for Nasopharyngeal Cancer in Non-Asian Patients: A Single-Center Experience"

_jcm, 2025, doi:10.3390/jcm14041177_

Round 1

Reviewer 1 Report

Comments and Suggestions for Authors

This manuscript investigates and presents a study on nasopharyngeal carcinoma (NPC) in a non-Asian cohort treated at a tertiary cancer center in Romania, which reports the highest NPC incidence in Europe. This is an intriguing study, and addressing the outlined limitations in future research could further enhance the impact and applicability of these findings.

Major concerns of this study include the need for additional clarification on its weaknesses, such as the retrospective, single-center design and the lack of routine EBV and HPV testing, which limit generalizability and comprehensiveness. Additionally, the conclusions regarding the study outcomes may be premature.

1.     EBV Testing: What is the rationale for not routinely conducting Epstein-Barr virus (EBV) testing, despite its established correlation with nasopharyngeal cancer (NPC)?

2.     In this retrospective investigation, how were biases, including selection bias and missing data, mitigated?

3.     Despite your reference to the possible involvement of human papillomavirus (HPV) in nasopharyngeal carcinoma (NPC), what was the rationale for excluding HPV testing from the analysis? Would the inclusion of this data elucidate its significance within the Romanian context?

4.     In follow-up , what is your level of confidence in detecting late recurrences or long-term toxicities? Are there intentions to prolong the follow-up for more extensive data?

5.     How were the variations in induction and concurrent chemotherapy regimens accounted for in the analysis of their effects on survival outcomes?

6.     Could you furnish more comprehensive data regarding the complete range of acute and late toxicities, encompassing grade 1 and grade 2 events? This may offer a more lucid comprehension of treatment tolerability.

7.     The study indicates a benefit from cumulative cisplatin doses of 100 mg/m²; nonetheless, most guidelines advocate for greater doses. Could you clarify the rationale for use lower doses and whether increased doses could enhance outcomes?

8.     In the absence of a control group or randomized comparison, how do you address potential confounders when evaluating the efficacy of various chemotherapy regimens?

I think should include EBV/HPV testing, larger datasets, extended follow-up, and regional risk analysis while benchmarking findings globally for better context.

Comments on the Quality of English Language

Long sentences reduce readability. Minor grammatical and typographical errors, such as "Instensity Modulated Radiotherapy," should be fixed. Consistency in terminology is good, but abbreviations should be defined at first usage and documented in a glossary. Simplifying complex sentences and standardizing punctuation will increase clarity and precision.

Author Response

Thank you for your time and patience to review my proposed article. Your comments were valuable. Here are my responses.

1.EBV Testing: What is the rationale for not routinely conducting Epstein-Barr virus (EBV) testing, despite its established correlation with nasopharyngeal cancer (NPC)?

Response: At the time of the study period (2014-2021) and unfortunately not even in the present EBV and HPV DNA testing are not reimbursed in our country. If we wish to do these tests that needs to be an out of pocket payment for the patients. Since at the present there is no recommended standard to apply different treatment strategy based on DNA testing or copy number we found it unethical to ask or oblige the patients to spend money on it. At the present we are looking for a sponsorship to do EBV DNA copy testing for a prospective cohort. EBER testing on pathology specimen is available only in certain pathology labs starting from last year.

2. In this retrospective investigation, how were biases, including selection bias and missing data, mitigated?

Answer: We included in the study all consecutive patients who were addressed to our clinic and excluded only the metastatic, pediatric and those sent for reirradiation. In this way we have in the study all the curative patients. In the timeframe of 2014-2018 our clinic was the only one providing IMRT technique for our region of the country so I think that the study population reflects well the outcome because the vast majority of new cases were sent to our clinic. Regarding attrition bias we have a rigorous followup schedule for the patients with very high participation rate. For the few patients not showing up we conducted telephone interviews and received all the date of death from the National Registry for the Evidence of the Population who provide a service for medical researchers by offering the exact date of death on request. 

3. Despite your reference to the possible involvement of human papillomavirus (HPV) in nasopharyngeal carcinoma (NPC), what was the rationale for excluding HPV testing from the analysis? Would the inclusion of this data elucidate its significance within the Romanian context?

HPV testing was not excluded on purpose, see my reply for comment 1. Testing HPV would bring valuable information regarding the unusual epidemiology observed for NPX for our country. We are working on a prospective cohort to get funding for comprehensive testing. 

4.  In follow-up , what is your level of confidence in detecting late recurrences or long-term toxicities? Are there intentions to prolong the follow-up for more extensive data?

My confidence to detect late recurrences is very high because our institutional protocol includes imaging 3-6 months in the first 3 years and at least 1 CT scan/ year and 2 clinical visits/ year from year 3 onward  with the possibility of performing PETCT in case of any suspicion. After 5 years of followup patient can opt to come to our clinic for yearly clinical visit and they are encouraged to present at any clinical sign or symptom. Since we are considered a reference center for treating head and neck cancer in the region late relapses outside the usual followup are sent for evaluation to us. In terms of late side effects we plan to establish a "Radiotherapy late effect clinic" to diagnose and manage late effects after 5-10 years from radiation. For this cohort we plan to do regular update on disease control and toxicity in the future. 

5. How were the variations in induction and concurrent chemotherapy regimens accounted for in the analysis of their effects on survival outcomes? 

Induction chemotherapy was prescribed for 72.7% of the patients. We need not to forget that 88.2 % were stage III and IVA disease which have standard indication for induction or adjuvant chemotherapy. Those not receiving induction were either stage I, II disease or cisplatin ineligible patients so comparison between induction vs no induction has no statistical meaning for the two group.  Regarding induction chemo type we know from the network meta-analysis by Choi et al that cisplatin-docetaxel, cisplatin-gemcitabine, and cisplatin-capecitabine regimes are the three most efficacious IC regimes based on the findings of studies comparing IC (different IC regime in each study) followed by CCRT vs CCRT alone and only a minor benefit was observed in favour of TPF regimen which was not clinically significant. (1) We analysed concomitant chemotherapy  Cisplatin weekly versus three weekly and found no difference in terms of disease control. The lack of concomitant Cisplatin however had negative impact as shown in Figure 2. 

6. Could you furnish more comprehensive data regarding the complete range of acute and late toxicities, encompassing grade 1 and grade 2 events? This may offer a more lucid comprehension of treatment tolerability. Yes- I will modify the toxicity data table and report separately G1 to G4.

7.  The study indicates a benefit from cumulative cisplatin doses of 100 mg/m²; nonetheless, most guidelines advocate for greater doses. Could you clarify the rationale for use lower doses and whether increased doses could enhance outcomes?

We have chosen the 100 mg/m2 dose because all patients with concomitant Cisplatin indication will receive at least one 100 mg/mp cycle or at least 2-3 weekly cycles (80-120 mg/mp) before toxicity would prevent the administration of further cycles in some patients. I agree that there are publications with higher doses of Cisplatin but there are also  studies reporting that after optimal induction chemotherapy with the use of IMRT as standard technique there might be a group of patients where lower or even omission of Cisplatin might be possible. (2,3) Omission in our study resulted in worse survival, that is why we wanted to see the impact of  a lower dose. 

8.     In the absence of a control group or randomized comparison, how do you address potential confounders when evaluating the efficacy of various chemotherapy regimens?

Since this is a retrospective study we don't have a control group and the scope was not to evaluate the efficacy of various chemotherapy regimens. We know from metaanalyses and large randomized studies the role and efficacy of each chemotherapy regimens. Our objective was to evaluate that in a non-Asian cohort, from a country with large incidence the survival results are similar or not to those reported in the literature mainly for Asian populations or other non-endemic low incidence countries.  

(1)H.C.-W. Choi, S.-K. Chan, K.-O. Lam, S.-Y. Chan, S.-C. Chau, D.L.-W. Kwong, et al. The most efficacious induction chemotherapy regimen for locoregionally advanced nasopharyngeal carcinoma: a network meta-analysisFront. Oncol., 11 (91) (2021) (2). Peng L, Chen JL, Zhu GL, Huang CL, Li JY, Ma J, Wen WP, Tang LL. Treatment effects of cumulative cisplatin dose during radiotherapy following induction chemotherapy in nasopharyngeal carcinoma: propensity score analyses. Ther Adv Med Oncol. 2020 Jun 25;12:1758835920937424. doi: 10.1177/1758835920937424. PMID: 32647541; PMCID: PMC7325541. (3). Dai J, Zhang B, Su Y, Pan Y, Ye Z, Cai R, Qin G, Kong X, Mo Y, Zhang R, Liu Z, Xie Y, Ruan X, Jiang W. Induction Chemotherapy Followed by Radiotherapy vs Chemoradiotherapy in Nasopharyngeal Carcinoma: A Randomized Clinical Trial. JAMA Oncol. 2024 Apr 1;10(4):456-463. doi: 10.1001/jamaoncol.2023.6552. 

Reviewer 2 Report

Comments and Suggestions for Authors

This article addresses a highly relevant topic for the scientific community, focusing on nasopharyngeal cancer (NPC) in non-Asian populations. The study is particularly important as it provides insights into a demographic group that is less frequently studied than endemic regions. By presenting survival data and treatment outcomes, the authors contribute valuable information that can guide clinical practice and improve outcomes for this population.

I have some questions and suggestions:

  • Why was this title chosen? The title suggests a multi-center study since the study is from a single center. I would recommend a more focused title to better reflect the scope of the work.
  • The article mentions that tumors were "histologically proven non-metastatic." Could you clarify this? Does "histologically proven" indicate that the cancer diagnosis was confirmed through microscopic examination of biopsy tissue samples and that no emboli were observed? Was immunohistochemistry for cytokeratin performed to rule out micrometastases? Were lymph nodes examined? Detailing the methods used to confirm the absence of metastasis would be beneficial. Was a standardized diagnostic protocol applied to ensure consistent staging across patients?
  • A stronger justification for why EBV testing and HPV co-testing were not routinely performed would address potential reader concerns about the completeness of the diagnostic data.
  • The discussion could be expanded to include how the study's findings might influence clinical guidelines or decision-making. This is particularly important for contexts with limited resources for advanced diagnostic tests, such as EBV or HPV screening.

Finally, I recommend a careful review of punctuation throughout the manuscript. There are several instances of misplaced spaces before periods.

This study significantly contributes to the understanding of NPC in non-Asian populations. Revisions to clarify the methods and emphasize clinical implications can further enhance its value to the scientific community.

Author Response

Thank you for your time and patience to review my article. Here are my responses to your comments. 

1. Why was this title chosen? The title suggests a multi-center study since the study is from a single center. I would recommend a more focused title to better reflect the scope of the work.

Answer: I have chosen this title because I wanted to outline the main objective- that to report survival data in a population in which Nasopharyngeal cancer is considered rare. The vast majority of our clinical knowledge comes from studies from south-east Asia and we tend to extrapolate those results to our non-Asian population. To clarify better the readers I will modify the title to Clinical Outcomes for Nasopharyngeal Cancer in non-Asian patients. A single-center experience.

2.The article mentions that tumors were "histologically proven non-metastatic." Could you clarify this? Does "histologically proven" indicate that the cancer diagnosis was confirmed through microscopic examination of biopsy tissue samples and that no emboli were observed? Was immunohistochemistry for cytokeratin performed to rule out micrometastases? Were lymph nodes examined? Detailing the methods used to confirm the absence of metastasis would be beneficial. Was a standardized diagnostic protocol applied to ensure consistent staging across patients?

Answer: All patients had biopsy from their primary tumour or cervical node. Metastatic status was not based on biopsy or any pathological finding but on imaging. Our institutional protocols include MRI of the head and neck, CT of the head and neck, thorax and abdomen and in selected cases with N3 disease or suspicious lesions on routine CT the addition of PET.  I will include this in the Materials and Methods section. 

3.A stronger justification for why EBV testing and HPV co-testing were not routinely performed would address potential reader concerns about the completeness of the diagnostic data.

At the time of the study period (2014-2021) and unfortunately not even in the present EBV and HPV DNA testing are not reimbursed in our country. If we wish to do these tests that needs to be an out of pocket payment for the patients. Since at the present there is no recommended standard to apply different treatment strategy based on DNA testing or copy number we found it unethical to ask or oblige the patients to spend money on it. At the present we are looking for a sponsorship to do EBV DNA copy testing for a prospective cohort. EBER testing on pathology specimen is available only in certain pathology labs starting from last year.

4. The discussion could be expanded to include how the study's findings might influence clinical guidelines or decision-making. This is particularly important for contexts with limited resources for advanced diagnostic tests, such as EBV or HPV screening.

I will include some paragraphs about this in the Discussion section.

Finally, I recommend a careful review of punctuation throughout the manuscript. There are several instances of misplaced spaces before periods.

Thank you. 

I will correct that.

Reviewer 3 Report

Comments and Suggestions for Authors

Manuscript ID: jcm-3446845

Title: Clinical outcomes for nasopharyngeal cancer in a non-Asian patient population

The paper titled, " Clinical outcomes for nasopharyngeal cancer in a non-Asian patient population ", is well-written and well-organized. 

The authors have systematically presented the study.

The global message of the manuscript is clear.

EBV prevalence and histological subtypes in the introductions are highlighted.  The results are supported by comprehensive survival data and Kaplan-Meier curves.

But there are some comments that needs to be addressed to the authors:

(1) For clarity, please consider specifying the study design (e.g., retrospective cohort study from Romania) in the title.

(2) The abstract could emphasize the novelty of the study, such as being the largest retrospective series of NPC in Romania/Europe - "Romania has the highest  incidence of NPC in Europe"

(3) Please clarify the rationale for the lack of routine EBV testing and its implications for clinical practice

(4) In the section on methods, the Authors described the cohort and treatment protocols, but the Authors should provide more information about the institutional protocols used for treatment. Were they entirely adapted from international guidelines or some domestic guidelines were also modified?

(5) Please discuss the lack of EBV and HPV testing in more depth—how might this impact the interpretation of the results?

(6) Additional details about the relapse and metastases or maybe prognostic factors for recurrence can be presented.

(7) The toxicity data could benefit from a deeper discussion.

(8) Strengthening the discussion around the impact of induction chemotherapy is needed.

(9) Please highlight future research directions, such as incorporating EBV/HPV testing and exploring molecular subtypes of NPC in the Romanian population.

(10) Please describe how the data of this paper could influence regional or international treatment guidelines.

(11) Please ensure all references are up-to-date and include recent studies (e.g., from 2023-2024) related to NPC (not only paper from 2024 from Romania). The reference list is not extensive and should include relevant primary studies and meta-analyses.

(12) Please simplify complex sentences for better readability in the methods and results sections.

(13) Please enhance the clarity of Kaplan-Meier curves and tables by ensuring consistent formatting and labeling.

(14) Maybe native speaker stylistic edits could improve fluency and precision.

Comments on the Quality of English Language

Maybe stylistic edits could improve fluency and precision.

Author Response

Thank you for your time and patience to review my proposed article. Your comments were valuable. Here are my responses.

(1)For clarity, please consider specifying the study design (e.g., retrospective cohort study from Romania) in the title.

 I have modified the title into: Clinical Outcomes for Nasopharyngeal Cancer in Non-Asian Patients. A single-center experience.

(2) The abstract could emphasize the novelty of the study, such as being the largest retrospective series of NPC in Romania/Europe - "Romania has the highest  incidence of NPC in Europe"

I have introduced in the abstract in the Conclusions section.

(3) Please clarify the rationale for the lack of routine EBV testing and its implications for clinical practice

At the time of the study period (2014-2021) and unfortunately not even in the present EBV and HPV DNA testing are not reimbursed in our country. If we wish to do these tests that needs to be an out of pocket payment for the patients. Since at the present there is no recommended standard to apply different treatment strategy based on DNA testing or copy number we found it unethical to ask or oblige the patients to spend money on it. At the present we are looking for a sponsorship to do EBV DNA copy testing for a prospective cohort. EBER testing on pathology specimen is available only in certain pathology labs starting from last year.

(4) In the section on methods, the Authors described the cohort and treatment protocols, but the Authors should provide more information about the institutional protocols used for treatment. Were they entirely adapted from international guidelines or some domestic guidelines were also modified?

The treatment protocols are entirely adapted from international guidelines in terms of radiotherapy dose, target volumes, induction or concomitant chemotherapy regimens and doses. The staging protocols are slightly  modified because of the selective use of PET/CT and no routine use of EBV and HPV testing. I clarify these in the reviewed manuscript.

(5) Please discuss the lack of EBV and HPV testing in more depth—how might this impact the interpretation of the results?

(6) Additional details about the relapse and metastases or maybe prognostic factors for recurrence can be presented.

(7) The toxicity data could benefit from a deeper discussion.

In the revised manuscript I included more detailed discussion about EBV and HPV impact and more relapse and toxicity data. 

(8) Strengthening the discussion around the impact of induction chemotherapy is needed.

Induction chemotherapy was prescribed for 72.7% of the patients. We need not to forget that 88.2 % were stage III and IVA disease which have standard indication for induction or adjuvant chemotherapy. Those not receiving induction were either stage I, II disease or cisplatin ineligible patients so comparison between induction vs no induction has no statistical meaning for the two group.  Regarding induction chemo type we know from the network meta-analysis by Choi et al that cisplatin-docetaxel, cisplatin-gemcitabine, and cisplatin-capecitabine regimes are the three most efficacious IC regimes based on the findings of studies comparing IC (different IC regime in each study) followed by CCRT vs CCRT alone and only a minor benefit was observed in favour of TPF regimen which was not clinically significant. I include some additional paragraphs on this subject.

(9) Please highlight future research directions, such as incorporating EBV/HPV testing and exploring molecular subtypes of NPC in the Romanian population.

 We are working on a prospective cohort to get funding for comprehensive testing. 

(10) Please describe how the data of this paper could influence regional or international treatment guidelines.

Because this is a retrospective study it can not influence at this moment treatment guidelines but it can stimulate regional collaboration to start multicenter clinical trials for non-Asian patients focusing on molecular profile and ideal combinations of chemotherapy and radiotherapy. 

 (11) Please ensure all references are up-to-date and include recent studies (e.g., from 2023-2024) related to NPC (not only paper from 2024 from Romania). The reference list is not extensive and should include relevant primary studies and meta-analyses.

Noted.

(12) Please simplify complex sentences for better readability in the methods and results sections.

Noted. 

(13) Please enhance the clarity of Kaplan-Meier curves and tables by ensuring consistent formatting and labeling.

Noted.

(14) Maybe native speaker stylistic edits could improve fluency and precision.

Noted

Round 2

Reviewer 1 Report

Comments and Suggestions for Authors

Accept